# Association of avian biodiversity and West Nile Virus circulation in *Culex* mosquitoes in Emilia-Romagna, Italy

Yiran Wang[1,2]*, Mattia Calzolari[3], Gianpiero Calvi[4], Victoria M. Cox[1], Paola Angelini[5], Michele Dottori[3], William Wint[6], Sally Jahn[1], Giovanni Marini[7], Ilaria Dorigatti[1]

**1** MRC Centre for Global Infectious Disease Analysis, School of Public Health, Imperial College London, London, United Kingdom, **2** Grantham Research Institute on Climate Change and the Environment, Imperial College London, London, United Kingdom, **3** Istituto Zooprofilattico Sperimentale della Lombardia e dell'Emilia Romagna "Bruno Ubertini", Brescia, Italy, **4** Studio Pteryx, Basiano, Italy, **5** Settore Prevenzione collettiva e Sanità pubblica, Regione Emilia-Romagna, Bologna, Italy, **6** Research and Innovation Centre, Fondazione Edmund Mach, San Michele all'Adige, Italy, **7** Research and Innovation Centre, Fondazione Edmund Mach, San Michele all'Adige, Tennessee, Italy

* yiran.wang21@imperial.ac.uk

## Abstract

### Background

West Nile Virus (WNV) is a zoonotic arbovirus maintained in a transmission cycle between *Culex* mosquitoes and birds, occasionally spilling over into humans. The impact of avian biodiversity on WNV circulation remains debated, with studies reporting both negative and positive correlations (dilution and amplification effects respectively) across different settings. In Europe, this relationship remains largely unexplored, particularly in regions with high WNV transmission, such as Emilia-Romagna in Northern Italy.

### Methods

We explored the association between avian biodiversity and WNV circulation in *Culex* mosquitoes in Emilia-Romagna using 11 years (2013–2023) of entomological surveillance data paired with two avian data sources. We calculated avian biodiversity indices (Shannon's, Simpson's, and Chao2) from observation records from the Farmland Bird Index project and applied linear regression models to assess their relationship with WNV detection frequency. Moreover, we used Bayesian spatiotemporal regression models and gridded weekly avian abundance estimates from the eBird project to analyse the associations between avian species richness indices and WNV transmission risk quantified by vector index (VI) at 68 geolocated mosquito traps across the region.

**Data availability statement:** The code and processed data that support the findings of this study and allow reproduction of the results are publicly available in the GitHub repository at https://github.com/mrc-ide/WNV_biodiversity_EM.

**Funding:** The WNV data were collected as part of the activities conducted by the Emilia-Romagna Regional Plan for Surveillance and Control of Arboviruses. The Bird observation data were collected by Lipu (Lega Italiana Protezione Uccelli and Italian BirdLife partner) as part of the Farmland Bird Index project, funded by the Ministry of Agriculture, Food Sovereignty and Forestry through the NRN 2014/2022 programme, EAFRD - European Agricultural Fund for Rural Development. YW acknowledges funding from the NERC Grantham Institute SSCP DTP (grant number NE/S007415/1). YW, VC, SJ and ID acknowledge funding from the MRC Centre for Global Infectious Disease Analysis (reference MR/X020258/1), funded by the UK Medical Research Council (MRC). This UK funded award is carried out in the frame of the Global Health EDCTP3 Joint Undertaking. ID and SJ acknowledge research funding from Welcome Trust (grant 213494/Z/18/Z and 228185/Z/23/Z, 226727/Z/22/Z). The funders had no role in study design, data collection and analysis, decision to publish, or preparation of the manuscript.

**Competing interests:** The authors have declared that no competing interests exist.

## Results

We observed consistent negative associations between WNV detection frequency in the *Culex* population and avian biodiversity indices, supporting the dilution effect hypothesis (DEH). We found that non-passerine species richness was negatively associated with VI while passerine species richness showed a positive association after adjusting for covariates and spatial random effects. These findings suggest that passerines may amplify WNV transmission, whereas the presence of non-passerine species is associated with reductions in WNV circulation.

## Significance

This study provides the first empirical evidence supporting the DEH for WNV in Europe. These findings have important implications for biodiversity conservation and integrated public health surveillance activities across Europe.

## Author summary

West Nile Virus (WNV) circulates primarily between *Culex* mosquitoes and birds, yet the association between avian biodiversity and WNV transmission remains debated. Focusing on Emilia-Romagna, Italy, one of Europe's WNV hotspots, we analysed 11 years (2013–2023) of entomological surveillance data alongside two avian datasets: bird census data from the Farmland Bird Index project and abundance estimates from the citizen-science project eBird. These datasets allowed us to test the dilution effect hypothesis (DEH) at both broad and fine spatiotemporal resolutions. Our findings support the DEH for WNV, showing that higher avian biodiversity is associated with reduced WNV activity. Additionally, we found that non-passerine species richness is linked to lower WNV transmission risk, while passerine species richness is associated with higher risk. These results point towards the potential amplifying role of passerine species and the protective role of non-passerine species in WNV transmission dynamics, with implications for species conservation, public health practice, and future modelling studies.

## Introduction

West Nile Virus (WNV) is a single-stranded, positive-sense RNA virus of the genus *Flavivirus* (family *Flaviviridae*), first isolated in Uganda in 1937 [1,2]. It is a zoonotic arbovirus primarily transmitted by *Culex* mosquitoes and maintained through a mosquito-bird-mosquito transmission cycle, with birds serving as main reservoir hosts and humans and horses as dead-end hosts [3]. The relevance of a bird species in WNV transmission primarily depends on the rate of contact with mosquito vectors and its reservoir competence. The rate of contact is largely reflected in mosquito

feeding patterns observed in the field, shaped by a combination of mosquitoes' intrinsic host preference and ecological factors such as host abundance, behaviour, and host-vector interactions [4,5]. Blood-meal analyses revealed that *Culex* mosquitoes predominantly feed on *Passeriformes* birds [6,7]. In North America, American Robin (*Turdus migratorius*), House Sparrow (*Passer domesticus*), and House Finch (*Haemorhous mexicanus*) are the most common blood-meal sources [6,8,9]. In Europe, common Blackbirds (*Turdus merula*) and House Sparrows are the primary targets [10–12], with Magpies (*Pica pica*) also frequently fed upon in Italy [13]. Reservoir competence refers to a host's ability to infect feeding mosquitoes and is determined by the magnitude and duration of infectious viremia following WNV infection. Laboratory studies measuring viremia profiles after experimental infections identified *Passeriformes*, particularly from the *Corvidae*, *Fringillidae*, and *Passeridae* families as highly competent WNV hosts [14]. Consequently, passerine birds are generally regarded as the optimal hosts for WNV transmission.

While WNV human infections are mostly asymptomatic (~80%) or cause mild, self-limiting flu-like symptoms such as headaches and fever (~20%), less than 1% of the WNV infections progress to severe West Nile neuroinvasive disease (WNND) which can be fatal, primarily affecting older individuals [15]. Currently, no targeted therapeutics or vaccines are available for WNV infection in humans. In recent years, WNV has caused several outbreaks in Europe [16–18], including a large epidemic in 2018, in which more human cases than the combined total of the previous seven years were reported [16], with a significant number of WNND cases and fatalities. While several modelling studies have investigated the role of environmental drivers and climatic anomalies in driving WNV transmission [19–21], less progress has been made in investigating the role of avian biodiversity in WNV transmission.

Biodiversity is defined as the natural variety and variability among living organisms, the ecological complexes they inhabit, and their interactions with one another and with the physical environment [22]. It is typically characterized by species richness, species evenness, and community composition [23]. Increasing evidence suggests that biodiversity might play a pivotal role in influencing the transmission of infectious diseases, particularly vector-borne zoonoses [24]. In 2000, Ostfeld and Keesing first proposed the dilution effect hypothesis (DEH), which suggests that higher biodiversity within a host community is associated with reduced infection rates in vectors and ultimately lowers the risk of pathogen transmission to humans [24]. Over the past two decades, the DEH has been demonstrated in multiple vector-borne zoonotic disease systems, such as Lyme disease [25,26], Tick-borne Encephalitis [27], Cutaneous leishmaniasis [28], and, more recently, Usutu Virus [29].

In the case of WNV, there is no consensus on the role of avian host biodiversity in transmission, with findings supporting both dilution and amplification effects. Several studies in the United States support the DEH, suggesting that greater avian biodiversity is associated with reduced WNV transmission risk [30–33]. In contrast, more recent studies in North America and Europe have identified amplification effects, where greater avian biodiversity is linked to increased WNV circulation [34,35]. These conflicting findings and the limited number of European studies underscore the need to investigate the location-specific relationship between avian biodiversity and WNV transmission, which is crucial for understanding the local disease ecology and developing effective prevention and control strategies that align with the One Health framework. Primarily due to the difficulty in obtaining reliable data on the bird community, such analyses remain largely underexplored, including in high WNV transmission regions of Europe, such as Emilia-Romagna in Northern Italy [36].

In this study, we investigate the effect of avian biodiversity on WNV circulation in *Culex* mosquitoes in Emilia-Romagna, Italy, by analysing 11 years of entomological surveillance data from 2013 to 2023 and two ornithological data sources: local bird census data from the Farmland Bird Index project and weekly species abundance estimates from the eBird project. We test the effect of avian biodiversity on WNV circulation as measured by alternative metrics at different spatial and temporal resolutions, adjusting for meteorological and land-use confounders. The insights generated by the analyses presented in this study have the potential to inform bird surveillance programs and guide the integration of biodiversity conservation into public health strategies, both within Emilia-Romagna and in broader contexts.

## Methods

### Study area

Emilia-Romagna region covers around 22,450 km$^2$ and has a resident population of about 4.47 million people [37]. From 2009, Emilia-Romagna adopted the national surveillance arbovirus plan locally and implemented an enhanced integrated surveillance program targeting mosquitoes, animals and humans which has been continuously refined over time and is still active to date [37,38]. Surveillance activities focus on an approximately 11,000 km$^2$ area located in the Po Valley plain, where over 90% of the region's residents live. This area is characterized by optimal ecological conditions for WNV circulation [37].

### Mosquito surveillance data

Mosquito surveillance in Emilia-Romagna was initiated after the first human WNV case was reported in the region in 2008 [39] and since then has been conducted annually during the summer months (June–October). Initially, in 2009, mosquito collections were performed at both fixed and temporary stations, with sampling frequencies ranging from weekly to monthly. From 2010 onward, the collection process was standardized, with fixed geo-referenced stations sampled every two weeks [37]. The surveillance network covers comprehensively the regional plain area using a grid of 11 km × 11 km cells, with traps strategically located to optimize mosquito collection [37]. Each trap is activated for a one night per sampling event, after which captured mosquitoes are counted, identified to species, and pooled based on date, location, and species, with a maximum of 200 individuals per pool. The pooled samples are subsequently tested for WNV presence and lineage using real-time polymerase chain reaction (RT-PCR) [37].

The mosquito data used in this study were obtained from 68 traps that were active throughout 2013–2023, the vast majority of which were $CO_2$-baited traps. 14 gravid traps were deployed only during the first five seasons (2013–2017) and were subsequently replaced by $CO_2$-baited traps at the same locations. Collections from these gravid traps accounted for 8.6% of all mosquito pool entries in the dataset.

### Farmland Bird Index data analysis

**Bird observation data.** The bird observation data used in this analysis were collated by Lipu (Italian BirdLife partner) within the Farmland Bird Index project funded by the Ministry of Agriculture, Food Sovereignty and Forestry through the European Agricultural Fund for Rural Development [40,41]. Data collection is based on a stratified sampling design with single-visit point counts [41]. Italy was divided into 10 km × 10 km squares (UTM projection), and within each square, point counts were conducted in 15 of the 100 1 km × 1 km cells, which serve as the sampling units for this study. Observations were carried out using unlimited-distance point counts [42], where trained observers recorded bird species and abundance through both visual and auditory detections, regardless of the distance from the observation point, in a 10-minute count period. These surveys were conducted annually between the beginning of May and the end of June (beginning of July in mountain regions) during morning hours [41]. Further methodological details of the project are available on the website: www.reterurale.it/farmlandbirdindex.

**Mosquito and bird data integration.** 68 mosquito traps that were active during all surveillance seasons from 2013 to 2023 were included in this study. Given that *Culex* mosquitoes have been documented to fly up to 2 miles (~ 3.2 km) [43], bird observation records were extracted from sampling units located within a 3-km buffer of the selected mosquito traps to ensure that the bird observations represent the avian communities most likely to be encountered by local mosquitoes (Fig 1). WNV activity at each trap was quantified as the number of years with at least one mosquito pool testing positive for WNV. This metric was selected because it showed a strong positive correlation with the proportion of mosquito pools testing positive during the study period (Pearson's r = 0.77, 95% CI: 0.65–0.85, p < 0.001), suggesting that WNV detection frequency serves as a reasonable indicator of local WNV circulation intensity.

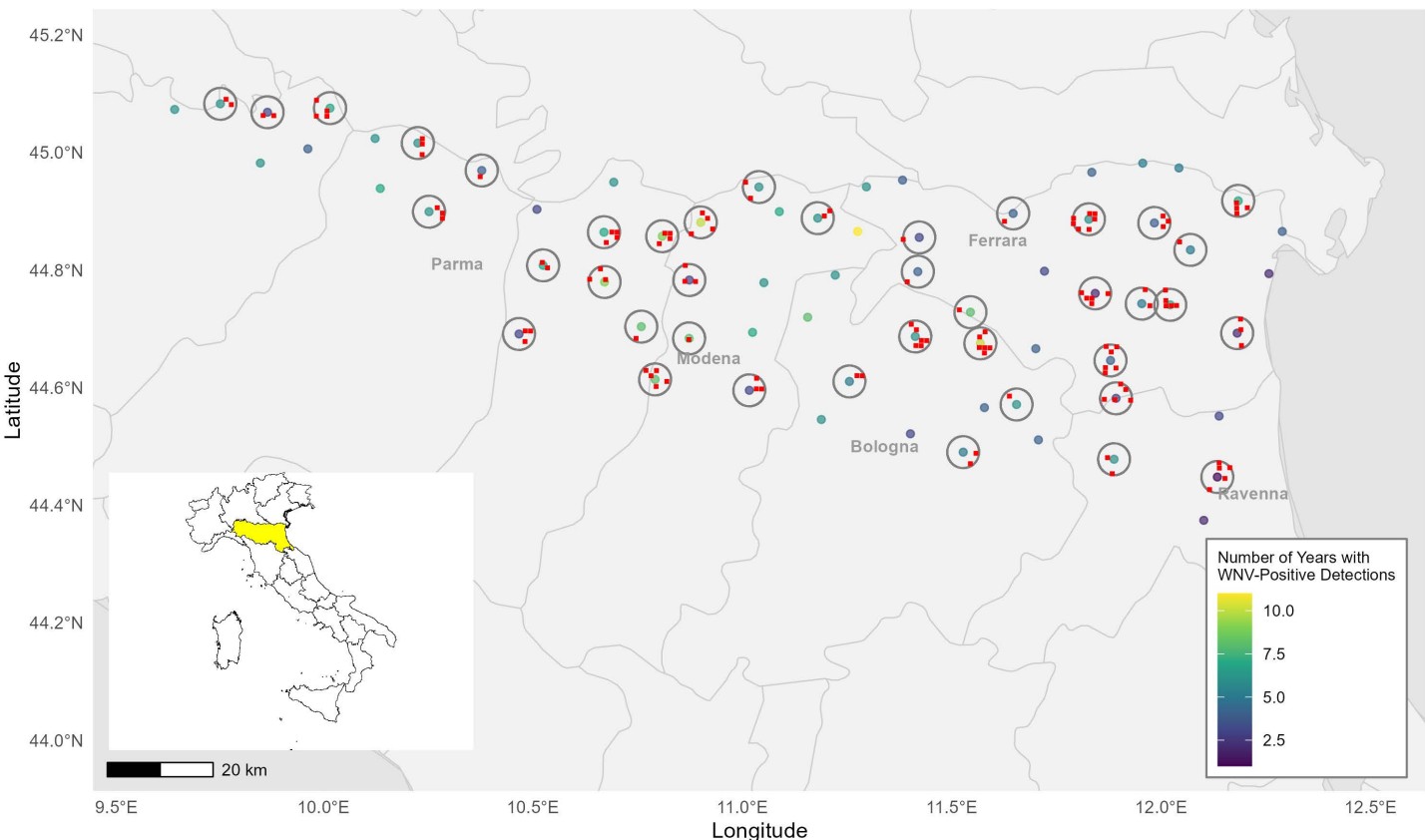

**Fig 1. Map of WNV mosquito surveillance traps active from 2013 to 2023 (points, color-coded from purple to yellow by WNV detection frequency, classified by the number of years with at least one mosquito pool testing WNV positive) in Emilia-Romagna (highlighted yellow on the map of Italy), with corresponding buffer areas (grey circles) and bird observational stations (red squares).** Some traps do not have associated buffer areas because no matched bird observation data were available. Base map from Natural Earth (public domain). Country borders (Admin 0 – Countries) and regional borders (Admin 1 – States, provinces) at 1:50m scale were obtained from Natural Earth 50m cultural vectors (https://www. naturalearthdata.com/downloads/50m-cultural-vectors/).

**Group classification and rarefaction.** Mosquito traps were grouped according to the number of years in which at least one WNV-positive mosquito pool was detected. Bird observation records within the buffer zones surrounding these traps were then compiled for each group. In the main analysis, we included 9 mosquito traps that were relocated to nearby sites within the same surveillance grid during the study period alongside the 68 traps that were active throughout the entire study period. A sensitivity analysis was conducted using only data from the 68 active traps (see S2 Text).

To account for differences in the number of observations across groups when calculating biodiversity indicators, sample-based rarefaction was applied [44,45]. This method involves repeatedly drawing $n$ samples without replacement from each assemblage (i.e., set of samples), where $n$ is the smallest number of samples among all assemblages [45]. The mean biodiversity indices calculated across these repeated subsamples represent the expected values for each assemblage. In this analysis, rarefaction was performed with 1,000 iterations.

The number of observations within each WNV detection frequency group is shown in Fig 2a and minimum threshold for reliable sample-based rarefaction was considered to be 20 [46]. In the Exclusion-Based Rarefaction approach, rarefaction was performed with a minimum of 31 observations across the remaining groups (Fig 2b). In the Combination-Based Rarefaction approach, consecutive groups were combined, and rarefaction was conducted using 75 subsamples (Fig 2c).

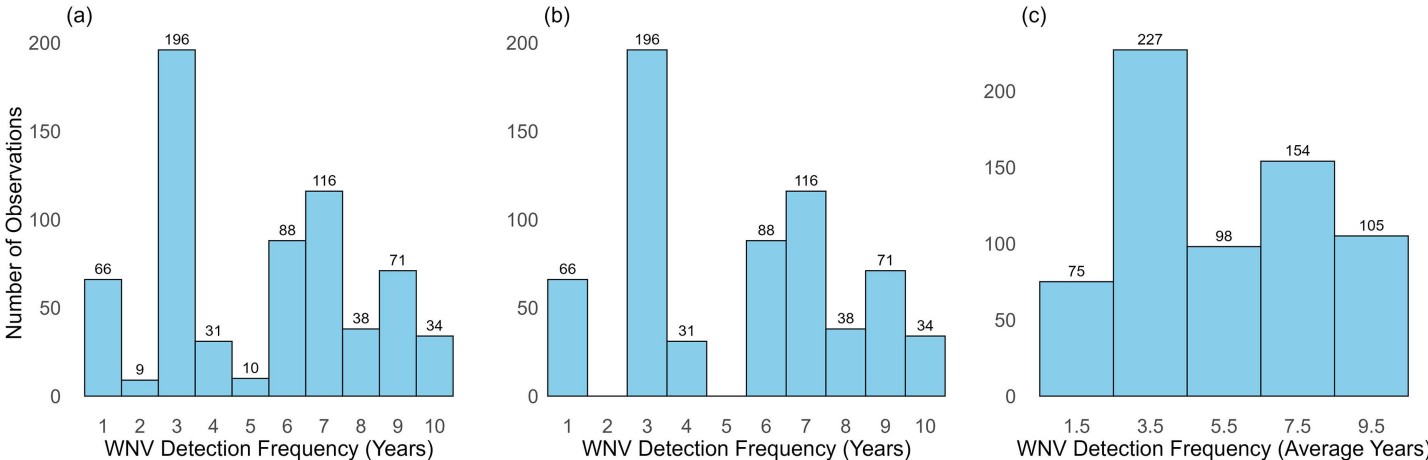

**Fig 2. Number of bird observations categorized by WNV detection frequency, based on the number of years with at least one WNV-positive mosquito pool. (a)** Classification using the original group definitions. **(b)** Classification after excluding groups with fewer than 20 observations. **(c)** Classification with the original two consecutive groups (as shown in panel a) combined.

**Biodiversity indicators.** For each group, three rarefied biodiversity indicators were calculated: Shannon's and Simpson's diversity indices (both increase with diversity) [47,48], and Chao2 Index, a non-parametric species richness estimator accounting for unseen species based on the frequency of rare ones, which is particularly useful in situations where not all species are observed due to sampling limitations [49]. The Shannon and Simpson diversity indices were calculated using the *vegan* R package [50]. Equations and parameter details are provided in the Supplementary Information (S1 Text).

**Statistical analysis.** Simple linear regressions were performed to explore the relationships between the three biodiversity indicators and WNV detection frequency in *Culex* mosquitoes.

## eBird data analysis

**eBird data.** This analysis used data from the eBird status and trends project developed by Cornell Lab of Ornithology [51,52], which applies machine learning models to estimate distributions, relative abundance, and population trends at high spatial and temporal resolutions across the full annual cycle for over 10,000 bird species. These estimates are generated by analysing the relationships between bird observations from eBird, a large-scale citizen science project, and habitat variations while accounting for biases and noise inherent in community science data, such as variability in bird observer behaviour and effort.

Weekly relative abundance data with a 3 km × 3 km spatial resolution for May to September (calendar weeks 18–39) were downloaded from the eBird website (https://science.ebird.org/en/status-and-trends) for bird species classified within the Emilia-Romagna region. eBird provides two versions of estimates: the 2021 version (based on data from 2007–2021) for some species and the 2022 version (based on data from 2008–2022) for others. Given that the population distribution of one species is largely consistent across years and only the presence/absence status is of interest, we downloaded data for species with no missing values during the study period, which reflect seasonal trends in the region.

Of the 308 bird species identified in eBird within the Emilia-Romagna region, 148 had complete relative abundance data during the study period and were included in the analysis (see S3 Table). The 148 species reported in eBird encompassed 90% of the top 30 species reported in the Farmland Bird Index project within a 10 km buffer of the mosquito surveillance traps from 2013 to 2023, and 80% of the top 50 species ranked by detection frequency. Weekly abundance

estimates for the 148 species were transformed into binary presence/absence data and aggregated into weekly species richness values, including overall bird species richness as well as passerine and non-passerine richness. In a sensitivity analysis, we explored the role of migratory birds in WNV circulation using the same analytical framework described above and weekly species richness estimates calculated exclusively for species classified as fully migratory in the study region.

**Environmental and meteorological data.** Meteorological data, including daily measurements of mean, minimum, and maximum temperature, precipitation, relative humidity, solar radiation, leaf wetness hours, and evapotranspiration, were obtained from the ERG5 dataset provided by the Regional Agency for Prevention, Environment, and Energy of Emilia-Romagna, Italy (ARPAE) [53,54] for the entire study period from 2013 to 2023. For each meteorological variable, weekly averages or cumulative values were computed and then averaged over the study period to obtain seasonal trends.

Land use proportions around mosquito traps were derived from the 2012 Corine Land Cover (CLC) dataset provided by the European Environment Agency, which classifies land cover into five categories: artificial surfaces (CLC1), agricultural areas (CLC2), forests and semi-natural areas (CLC3), wetlands (CLC4), and water bodies (CLC5) [55]. For each trap location, land use types within a 3 km buffer were identified by extracting CLC data from shapefiles provided by the Institute for Environmental Protection and Research (ISPRA) using QGIS (www.qgis.org). The proportion of each land cover class within each mosquito trap's buffer zone was then calculated.

**Vector index.** For the 68 mosquito traps that were active throughout each surveillance season and consistently sampled during calendar weeks 18–39 from 2013 to 2023, weekly mosquito surveillance records were aggregated across all years. For each trap and week, we utilised the total number of mosquito pools tested for WNV, the number of WNV-positive pools, their unique pool sizes, and mosquito abundance per trap-night to compute the Vector Index (VI). The VI represents the estimated number of WNV-infected mosquitoes collected per trap-night and is derived from:

$$VI = N \times P \tag{1}$$

where $N$ refers to the mosquito abundance per trap-night, and $P$ denotes the estimated infection rate [56]. Infection rates were obtained using the *PooledInfRate* R package [57]. A key advantage of using the VI as the response variable for the spatiotemporal analysis is in the integration of vector species infection rate with vector abundance data, which provides a composite measure of WNV transmission risk [56].

**Spatiotemporal analysis.** A Bayesian spatiotemporal model [58] was adapted to examine the relationships between WNV transmission risk and bird species richness, adjusting for environmental covariates in a seasonal context using the Integrated Nested Laplace Approximation - Stochastic Partial Differential Equation (INLA-SPDE) approach implemented in R-INLA (www.r-inla.org) [59,60]. This approach was chosen because it efficiently models spatially structured ecological processes from irregularly spaced sampling locations, as in the mosquito surveillance network analysed.

WNV transmission risk, as measured by VI, was assumed to be a continuous, spatially correlated process across the region. This assumption was validated through spatial autocorrelation analysis, which revealed significant clustering of VI (see S3 Text). The spatial process was represented using a discretized Gaussian Markov random field (GMRF) on a triangulated mesh [61,62]. The model combines the spatial field (a random effect) with the avian, meteorological, and land use variables (fixed effect covariates) to reconstruct the observed patterns in mosquito WNV circulation. Details on the mesh and prior distributions for the GMRF parameters are provided in S4 Text. Analyses were conducted using the *R-INLA* package (version 24.05.10).

A total of 87 initial variables were included in the analysis, including the bird species richness (overall, passerine, non-passerine), and meteorological and land use covariates. Species richness and meteorological variables were extracted with up to 3-week lags to account for lag effects. To identify key predictor variables, we employed Spike and Slab, a Bayesian variable selection method [63] performed using the *spikeslab* R package [64]. Pairwise correlations among the selected variables were then calculated, and highly correlated variables were excluded to mitigate

multicollinearity. For pairs of variables within the same category (e.g., temperature-related variables) with a correlation coefficient exceeding 0.6, the variable with the higher Bayesian model average (BMA) value was retained. For pairs of variables from different categories (e.g., temperature and precipitation), priority was given to removing compound variables (e.g., evapotranspiration).

Following this procedure, six variables ($X_{\{ij\},t}$, j = 1,..., 6)—namely, passerine species richness at zero-lag, non-passerine species richness at zero-lag, weekly average temperature at a three-week lag, cumulative precipitation aggregated over lag weeks two to zero, weekly average solar radiation at a one-week lag, and land use type CLC1 (artificial surfaces)—were included as fixed effects in the final spatiotemporal model. Although the other three land use variables (CLC2, CLC3, CLC4) were initially retained alongside these six variables after the variable selection process, they were not statistically significant and were therefore excluded from the final model to avoid potential bias from including a non-informative covariate. Given that VI included a high proportion of zeros (70.12%), we applied a zero-inflated Poisson distribution to account for the two types of zeros: structural zeros (representing locations where no WNV-infected mosquitoes were expected) and sampling zeros (representing locations where no WNV-infected mosquitoes were detected due to sampling limitations). The model was specified as follows:

$$\Pr\left(y_{i,t}\right) = \pi_{i,\ t} + \left(1 - \pi_{i,t}\right) \cdot Poisson\left(y_{i,t} \mid \lambda_{i,t}\right) \tag{2}$$

where $y_{i,\ t}$ is the VI at trap $i$ in week $t$; $\pi_{i,t}$ represents the probability of a structural zero; and $\lambda_{i,\ t}$ is the expected VI from the model, which was defined as shown in eq. 2:

$$logit\left(\lambda_{i,\ t}\right) = \beta_0 + \sum_{j=1}^{6} \beta_j X_{\{ij\},t} + u_i \tag{3}$$

where $\beta_0$ is the intercept; $\beta_j$ is the regression coefficient for the predictor $X_{\{ij\},t}$ at trap location $i$ and week $t$; and $u_i$ is the spatial random effect at trap location $i$.

We also performed a sensitivity analysis using the same analytical framework but only including total species richness metrics to investigate the potential joint effect of the local bird community, without distinguishing passerine and non-passerine richness variables.

All analyses were performed using R version 4.4.0 [65]. The data and code used in this analysis are provided in: https://github.com/mrc-ide/WNV_biodiversity_EM.

## Results

### Farmland Bird Index data analysis

Among the 86 mosquito traps included in this study, including 68 traps that were active throughout all surveillance seasons from 2013 to 2023 and 9 traps that were relocated to nearby locations during the study period, 55 traps had matched bird observation records conducted across different times. These traps yielded a total of 659 bird observation records, encompassing 107 species and 26,608 individuals. Of the 107 species, 44 were passerines, accounting for 15,264 individuals. Among the 55 mosquito traps with associated bird observation records, WNV was detected in all traps in at least one year and up to a maximum of ten years. Across these 55 traps, 448 out of 20,981 tested mosquito pools were positive for WNV, 447 of which consisted of *Culex pipiens* specimens, with a single positive pool of *Culex modestus* mosquitoes. We considered all WNV-positive mosquito pools irrespective of lineage, despite WNV lineage 2 was predominant during the study period (i.e., only 9 pools were positive for lineage 1 and only 3 pools were positive for both lineage 1 and 2 during 2022–2023, compared to 150 pools that were positive only for lineage 2).

In the Exclusion-based Rarefaction approach, where WNV detection frequency groups with fewer than 20 bird observations were excluded and sample-based rarefaction was performed (Fig 2b), linear regression results indicated that both

rarefied Shannon's and Simpson's diversity indices were negatively associated with the number of years in which mosquito surveillance traps detected at least one WNV-positive pool (Shannon: β = –8.89, p = 0.005, $R^2$ = 0.75; Simpson: β = –59.04 , p = 0.006, $R^2$ = 0.75) (Fig 3a and 3b). This suggested that higher diversity in the bird community was associated with lower WNV circulation in *Culex* mosquitoes. While a negative trend was also observed between bird species richness (Chao2 index) and WNV circulation, this relationship was not statistically significant (β = –0.15, p = 0.376, $R^2$ = 0.13) (Fig 3c).

In the Combination-based Rarefaction approach, consecutive WNV detection frequency groups were combined and sample-based rarefaction was performed based on 75 subsamples (Fig 2c). Similarly to the Exclusion-based Rarefaction method, the results from simple linear regressions revealed negative associations between Shannon's and Simpson's diversity

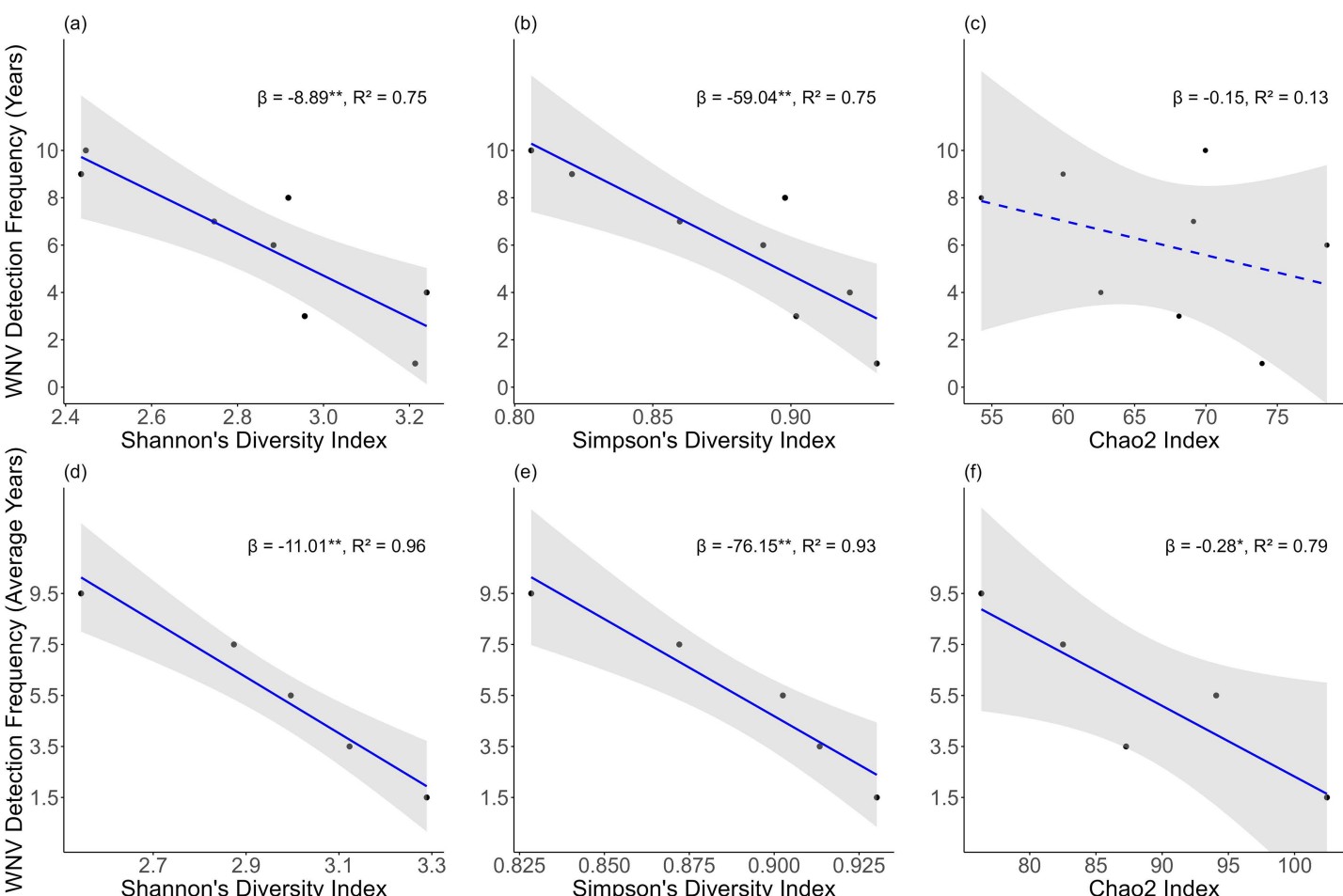

**Fig 3. Negative associations between rarefied bird community biodiversity indices and WNV detection frequency, measured as the number of years with at least one WNV-positive pool in the Exclusion-based Rarefaction approach (a-c) and the average number of years with at least one WNV-positive pool in the Combination-based Rarefaction approach(d-f), supporting the dilution effect hypothesis.** β and $R^2$ represent the coefficient and coefficient of determination from simple linear regression, respectively; * indicating a p-value (*p*) $p \leq 0.05$ and ** indicating $p \leq 0.01$. Solid lines were used for $p \leq 0.05$, a dashed line denotes a non-significant trend (*p* > 0.05). (a) Rarefied Shannon's Diversity Index versus WNV detection frequency (Exclusion-based Rarefaction). (b) Rarefied Simpson's Diversity Index versus WNV detection frequency (Exclusion-based Rarefaction). (c) Rarefied Chao2 Index versus WNV detection frequency (Exclusion-based Rarefaction). (d) Rarefied Shannon's Diversity Index versus WNV detection frequency (Combination-based Rarefaction). (e) Rarefied Simpson's Diversity Index versus WNV detection frequency (Combination-based Rarefaction). (f) Rarefied Chao2 Index versus WNV detection frequency (Combination-based Rarefaction).

indices and WNV detection frequency (Shannon: β = –11.01, p = 0.004, $R^2$ = 0.96; Simpson: β = –76.15, p = 0.008, $R^2$ = 0.93) (Fig 3d and 3e). Additionally, a negative relationship between the Chao2 index and the average number of years with at least one WNV-positive pool was observed, in this case significant (β = –0.28, p = 0.044, $R^2$ = 0.79), suggesting that higher species richness in the bird community was associated with lower WNV circulation in vectors (Fig 3f). The rarefied biodiversity indices (Shannon's, Simpson's, and Chao2) and corresponding regression outputs derived from the Exclusion-based and Combination-based Rarefaction approaches are presented in S1 and S2 Tables, respectively.

A sensitivity analysis, conducted using data only from the mosquito traps that were active throughout the entire study period (i.e., excluding the 9 traps that were relocated), showed consistent results across both rarefaction methods (see S4 Text).

**eBird data analysis**

The weekly species richness from May to September at 68 mosquito surveillance traps, active throughout all years, derived from the eBird weekly abundance estimates of 148 bird species in the Emilia-Romagna region, exhibits clear spatiotemporal heterogeneity (Fig 4). Spatially, species richness varies widely across trap locations each week. Temporally,

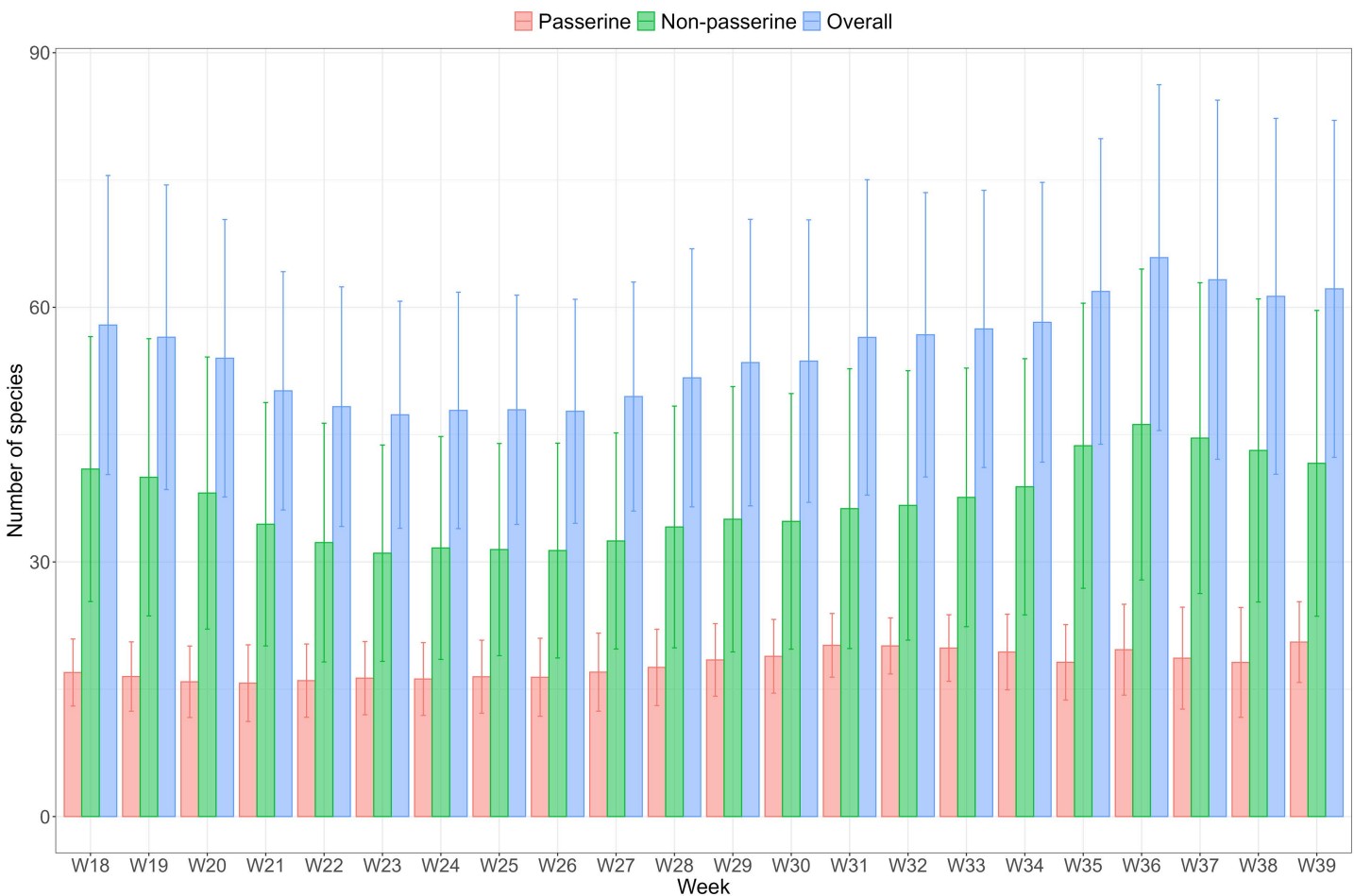

**Fig 4. Seasonal trends in weekly passerine (red), non-passerine (green), and overall bird species richness (blue) around 68 active mosquito surveillance traps in Emilia-Romagna, Italy (weeks 18–39, May to September), obtained from eBird abundance estimates.** The bars represent the mean values, with error bars indicating the 95% confidence intervals.

species richness is higher in correspondence with the migration periods, when several non-breeding species stopover in the study area for refuelling. At the beginning of the study period in May, the pre-breeding migration is at its final stage, which translates into a slight decline of species richness. From June to July, richness remains relatively stable, coinciding with the end of the breeding season, and then increases again between August and September, due to postnuptial migration. Bird species were stratified into passerines or non-passerines given that passerines are generally considered competent hosts that can amplify WNV transmission, whereas non-passerines are considered less competent and may dilute its transmission. Throughout the season, non-passerine species consistently outnumber passerine species around the mosquito surveillance traps. Among the 45 bird species classified as fully migratory (see S3 Table), only 12 were passerines.

Variables included in the spatiotemporal regression model and their associated estimated coefficients are presented in Table 1. The results indicate that non-passerine species richness was negatively associated with VI, after adjusting for covariates and the spatial random effect. This suggested that a greater presence of non-passerine species within the bird community was linked to reduced WNV transmission risk. In contrast, passerine species richness was positively associated with VI, indicating that a higher presence of passerine species was correlated with increased WNV circulation in the vector population. Additionally, the proportion of artificial surfaces and cumulative precipitation over the past two weeks were negatively associated with WNV transmission risk, while the weekly average temperature at a three-week lag and weekly average solar radiation at a one-week lag were positively associated. To assess model fit, the mean absolute error (MAE) was calculated for both the final model and a baseline model containing only the intercept and spatial random effect. The MAE for the final model was 1249, compared to 1716 for the baseline model, indicating that the inclusion of species richness variables alongside environmental (presence of artificial surfaces) and meteorological (temperature, precipitation and solar radiation) fixed effects substantially improved model performance.

In the sensitivity analysis investigating the role of the overall richness of the bird community in WNV transmission risk, total species richness was not significantly associated with VI, as the 95% credible interval (CI) crossed zero (mean coefficient: 0.000, 95% CI: [-0.001, 0.001]; S4 Table).

Further analysis focusing on migratory birds revealed that the species richness of fully migratory birds, with a three-week lag, was significantly negatively associated with VI, after adjusting for temperature, precipitation, agricultural areas, and the spatial random effect (mean coefficient: -0.034, 95% CI: [-0.035, -0.033]; S5 Table).

## Discussion

Our findings support the DEH of WNV transmission in Emilia-Romagna, Italy, and points toward the potential role of passerine birds in enhancing WNV circulation and non-passerine birds in mitigating its transmission.

In the Farmland Bird Index data analysis, Shannon's and Simpson's diversity indices were significantly lower around mosquito surveillance traps with increased WNV detection frequency. These two indices reflect the relative proportions of different species in the community, with lower values indicating simpler communities dominated by fewer species. This

**Table 1. Summary of fixed effects from Bayesian spatiotemporal regression for Vector Index representing WNV transmission risk in Emilia-Romagna: posterior mean and 95% credible intervals (CrI) for predictor variables.**

| Variables | Mean | 2.5% CrI | 97.5% CrI |
|---|---|---|---|
| Non-passerine species richness (lag 0) | -0.006 | -0.007 | -0.005 |
| Passerine species richness (lag 0) | 0.080 | 0.078 | 0.081 |
| CLC1 (Artificial surfaces) | -1.068 | -2.068 | -0.081 |
| Weekly Average Temperature (lag 3) | 0.095 | 0.093 | 0.098 |
| Cumulative Precipitation (lag 0–2) | -0.007 | -0.008 | -0.007 |
| Weekly Average Solar Radiation (lag 1) | 0.032 | 0.031 | 0.034 |

suggests that more diversified bird communities, with more complex trophic and ecosystem interactions, may be less permissive to WNV circulation, while simplified communities are likely more prone to viral transmission, which is consistent with findings from the USA [31–33]. Multiple studies have demonstrated negative associations between Shannon's diversity index and WNV transmission metrics, including human incidence [31], mosquito prevalence [32], and mortality in highly susceptible American crows [33]. The eBird analysis results are consistent with previous findings from Louisiana, USA, where non-passerine richness was negatively associated with WNV incidence in *Culex* mosquitoes [30]. Together, these findings corroborate the DEH of WNV transmission in Emilia-Romagna.

There are several potential mechanisms underlying the DEH in vector-borne disease systems, which could operate in concert in nature including: (i) transmission reduction (or encounter reduction), where additional species lower contact rates between vectors and competent hosts; (ii) susceptible host regulation, where interspecific competition limits highly susceptible hosts; (iii) vector regulation, where increased feeding on vectors by diverse bird species reduces vector density; and (iv) recovery augmentation, where additional species enhance the recovery of infected individuals by providing resources, such as acting as mutualists or prey [66]. Our findings align most closely with the encounter reduction theory, supported by the strong negative associations between Shannon's and Simpson's diversity indices and WNV detection frequency observed in the Farmland Bird Index analysis, as well as the significant negative relationship between the species richness of fully migratory bird species and VI in the eBird analysis. These results suggest that the introduction of additional bird species, possibly through migration and resulting in increased diversity of the bird community, may lower WNV circulation in *Culex* mosquitoes in our study site. Notably, Emilia-Romagna hosts a higher proportion of non-passerine species than passerine species during the WNV transmission season, with most fully migratory species being non-passerines.

Additionally, our eBird analysis provides statistical evidence linking greater passerine species richness to increased WNV transmission risk, which may imply a potential amplifying role of passerine birds in the spread of WNV. This finding is consistent with a study conducted in Southwest Spain, where a higher number of passerine species in the bird community was associated with increased prevalence of Usutu virus, a pathogen that can co-circulate with WNV and is also primarily transmitted by *Culex* mosquitoes [67]. This pattern may be explained by both higher reservoir competence of passerine species and/or their more frequent contact with mosquito vectors. While laboratory studies have demonstrated that many passerine species develop high viremia levels following WNV infection, these findings are largely based on North American species and the North American WNV lineage 1 [68], which may not be generalizable to other geographic contexts [69]. In Europe, a few studies have assessed the reservoir competence of local bird species using European WNV strains, with some but inconsistent evidence that European passerines exhibit high viremia levels as those observed in North American studies [70–72]. In contrast, a study analysing 150 European passerine species found that models incorporating life-history traits affecting their rate of contact with mosquitoes, such as body mass, nest height, and life stage, better explained variations in bird seroprevalence than models based solely on species identity. This suggests that variations in vector-host contact rates might play a vital role in shaping WNV transmission in Europe [73]. Therefore, the positive association observed between passerine species richness and VI in our study may reflect the frequent feeding of *Culex* mosquitoes on passerine birds, as supported by blood meal analyses from North America and Europe [6,8–13]. Nevertheless, further targeted studies are needed to better characterise the reservoir competence of European passerines and clarify their role in WNV transmission.

While we observed that non-passerine richness was negatively associated and passerine richness was positively associated with VI, in the eBird analysis we found no association between total bird species richness and VI, suggesting that total species richness alone may be insufficient for assessing biodiversity-disease relationships, as amplifying and dilution effects from hosts with different competence levels may counteract each other. Our results highlight the importance of accounting for community composition and of considering species-specific richness metrics when investigating associations with WNV risk estimates.

Our eBird analysis also revealed associations between meteorological and environmental factors and WNV transmission risk. The proportion of artificial surfaces was negatively associated, whereas weekly average temperature and solar radiation were positively associated with VI; these patterns are consistent with the literature [74–77]. A negative association was also observed between cumulative precipitation and VI, which may be explained by the influence of precipitation and adverse weather conditions on migratory bird stopover behaviour, thereby potentially altering bird community composition during migration periods [78].

Compared with previous studies which focused on single transmission seasons [30–32,35], this study analysed 11 years of mosquito surveillance and bird observation data. Additionally, we performed sample-based rarefaction to adjust for differences in sampling effort across trap locations and used the Chao2 index to estimate species richness, which accounts for unobserved rare species due to sampling limitations, providing a more accurate measure than simple species counts [30,31,35]. Moreover, in the eBird data analysis, we were able to examine the role of bird species richness in modulating the spatiotemporal dynamics of VI, which has been proven to be a reliable predictor for WNV human risk from previous studies [56], while comprehensively accounting for the confounding effects of meteorological and environmental factors, which is novel and possible due to the resolution of the eBird data.

A key limitation of the Farmland Bird Index data analysis is that it was conducted using aggregated trap data rather than individual trap-level observation records. While this ensured sufficient bird observations to derive biodiversity indices at each WNV detection frequency group, it limited the extent to which we could investigate year-specific associations and account for potential confounders such as meteorological and land-use factors at fine spatial resolution. Additionally, the calculation of Shannon's and Simpson's diversity indices from point-count detection data may be biased toward more conspicuous species, as these abundance-based indices do not account for species-specific detectability differences.

A potential drawback of the eBird data is that abundance estimates are based on the eBird citizen science project which by design does not systematically survey pre-defined locations and hence may introduce bias. Furthermore, these estimates reflect seasonal trends and do not capture interannual variations. Additionally, because eBird abundance estimates are relative and standardised within each species, they do not allow for direct comparisons between species, limiting our analysis to species richness rather than more comprehensive biodiversity metrics. As a result, while our findings show a positive association between passerine species richness and VI representing WNV transmission risk, it remains challenging to disentangle whether this reflects a genuine amplifying role or simply their higher availability as blood-meal sources, considering that *Passeriformes* is the largest avian order. Another limitation of the study is in the lack of sufficient data to identify which specific passerine groups or species may be responsible for any potential amplifying effect. These limitations are common to most bird data sources, given the difficulty of accounting for heterogeneity in detection and reporting rates across species. Finally, although we adjusted for environmental covariates that might affect WNV transmission risk, we could not incorporate ecological confounders such as the abundance of birds or mammals that sustain *Culex* populations over the season, due to the unavailability of data with sufficient temporal resolution for our study site.

It is also important to note that, as WNV lineage 2 was largely dominant in Emilia-Romagna during the study period according to mosquito surveillance, our results should be interpreted primarily as reflecting the association between avian biodiversity and the transmission dynamics of WNV lineage 2 in this region. They may not be directly generalisable to lineage 1 or to settings where lineage 1 predominates, given potential phenotypic differences between lineages, for instance in avian reservoir competence or in the vector competence of local *Culex* populations.

To date, the only study investigating the DEH in the case of WNV transmission in Europe, conducted in southern Spain in 2013, found a positive association between avian species richness and WNV seroprevalence in house sparrows [35], a known competent reservoir host [70]. The authors attributed this to *Culex perexiguus*, the primary WNV vector in the region, being more abundant in rural areas where avian biodiversity tends to be higher compared to urban settings. The divergent findings found in this study may reflect differences in data types (vector-based vs. avian serological

surveillance), primary vector species (*Culex pipiens* vs. *Culex perexiguus*), as well as different regional ecological contexts, emphasising the need for location-specific assessments of host-vector-pathogen dynamics.

The results presented in this study provide the first empirical evidence supporting the DEH for WNV transmission in a European setting. This analysis represents one of the most comprehensive studies examining avian biodiversity effects on vector-borne zoonosis transmission from 11 years of systematic entomological surveillance and multiple ornithological data sources, which has important implications for public health practice and biodiversity conservation. Our findings suggest that areas characterised by high passerine species richness or low overall avian biodiversity may experience higher WNV transmission risk. In practice, such areas could be prioritised for intensified mosquito surveillance, enhanced blood-donor and equine screening, strengthened risk communication, and targeted vector control to help prevent human and animal cases of WNV. The analytical approach developed in this study also highlights the value of generating high-resolution bird diversity and abundance maps, which can help improve our understanding of biodiversity's role in disease dynamics and demonstrate the beneficial role of avian biodiversity conservation in reducing the local long-term risk of WNV circulation in the bird, equids, and human populations.

Furthermore, our findings will inform the development of mechanistic models of WNV transmission, most of which currently consider general bird compartments or only competent bird hosts [79,80]. Finally, our study underscores the importance of coordination between disease surveillance and bird monitoring programs to facilitate targeted data collection for understanding the circulation patterns of WNV and other zoonotic diseases. Future studies can build on the results presented in this study to further explore the mechanisms underlying these associations and to assess how biodiversity conservation efforts may influence the environmental circulation of WNV.

## Supporting information

**S1 Text. Equations and parameter details of biodiversity indicators in the Farmland Bird Index data analysis.**
(DOCX)

**S2 Text. Sensitivity analysis for Farmland Bird Index data.**
(DOCX)

**S3 Text. Assessing spatial autocorrelation in Vector Index.**
(DOCX)

**S4 Text. Mesh construction and prior specifications of the INLA-SPDE model.**
(DOCX)

**S1 Table. Rarefied biodiversity indices (Shannon's, Simpson's, and Chao2) by eight WNV detection frequency groups and regression analysis results (Exclusion-based Rarefaction).** β refers to regression coefficients, 95% CI refers to 95% confidence intervals.
(DOCX)

**S2 Table. Rarefied biodiversity indices (Shannon's, Simpson's, and Chao2) by five WNV detection frequency groups and regression analysis results (Combination-based Rarefaction).** $\beta$ refers to regression coefficients, 95% CI refers to 95% confidence intervals.
(DOCX)

**S3 Table. Bird species in Emilia-Romagna, Italy with available weekly abundance estimates from May to September to download from the eBird website.** The entry '21' in column 'Version' indicates the 2021 version of eBird (based on data from 2007–2021) and '22' indicates the 2022 version (based on data from 2008–2022). The species classified as fully migratory are highlighted in bold.

(DOCX)

**S4 Table. Summary of fixed effects from Bayesian spatiotemporal regression examining the association between total bird species richness and Vector Index (VI) representing West Nile Virus (WNV) transmission risk in *Culex* mosquitoes: posterior means and 95% credible intervals (CrIs) for predictor variables.**
(DOCX)

**S5 Table. Summary of fixed effects from Bayesian spatiotemporal regression examining the association between fully migratory bird species richness and Vector Index (VI) representing West Nile Virus (WNV) transmission risk in *Culex* mosquitoes: posterior means and 95% credible intervals (CrIs) for predictor variables.**
(DOCX)

## Acknowledgments

We are grateful to all the observers involved in the bird census.

## Author contributions

**Conceptualization:** Yiran Wang, Giovanni Marini, Ilaria Dorigatti.

**Data curation:** Yiran Wang.

**Formal analysis:** Yiran Wang.

**Funding acquisition:** Yiran Wang, Ilaria Dorigatti.

**Investigation:** Yiran Wang, Mattia Calzolari, Gianpiero Calvi, Victoria M. Cox, Paola Angelini, Michele Dottori, William Wint, Sally Jahn.

**Methodology:** Yiran Wang, Mattia Calzolari, Gianpiero Calvi, Victoria M. Cox, William Wint, Giovanni Marini, Ilaria Dorigatti.

**Project administration:** Yiran Wang.

**Resources:** Mattia Calzolari, Gianpiero Calvi.

**Software:** Yiran Wang, Victoria M. Cox.

**Supervision:** Giovanni Marini, Ilaria Dorigatti.

**Validation:** Victoria M. Cox.

**Visualization:** Yiran Wang.

**Writing – original draft:** Yiran Wang.

**Writing – review & editing:** Yiran Wang, Mattia Calzolari, Gianpiero Calvi, Victoria M. Cox, Paola Angelini, Michele Dottori, William Wint, Sally Jahn, Giovanni Marini, Ilaria Dorigatti.

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
