## [Decision Letter · Decision Letter 0]

4 Nov 2025

PNTD-D-25-01508

Association of avian biodiversity and West Nile Virus circulation in *Culex* mosquitoes in Emilia-Romagna, Italy

Dear Dr. Wang,

Thank you for submitting your manuscript to PLOS Neglected Tropical Diseases. After careful consideration, we feel that it has merit but does not fully meet PLOS Neglected Tropical Diseases's publication criteria as it currently stands. Therefore, we invite you to submit a revised version of the manuscript that addresses the points raised during the review process.

Please submit your revised manuscript within 60 days Jan 03 2026 11:59PM. If you will need more time than this to complete your revisions, please reply to this message or contact the journal office at plosntds@plos.org. Please include the following items when submitting your revised manuscript:

We look forward to receiving your revised manuscript.

Kind regards,

Ran Wang, M.D.

Academic Editor

Elvina Viennet

Section Editor

Shaden Kamhawi

co-Editor-in-Chief

Paul Brindley

co-Editor-in-Chief

**Journal Requirements:**

1) Some material included in your submission may be copyrighted. According to PLOSu2019s copyright policy, authors who use figures or other material (e.g., graphics, clipart, maps) from another author or copyright holder must demonstrate or obtain permission to publish this material under the Creative Commons Attribution 4.0 International (CC BY 4.0) License used by PLOS journals. Please closely review the details of PLOSu2019s copyright requirements here: PLOS Licenses and Copyright. If you need to request permissions from a copyright holder, you may use PLOS's Copyright Content Permission form.

Potential Copyright Issues:

i) Figure 1. Please (a) provide a direct link to the base layer of the map (i.e., the country or region border shape) and ensure this is also included in the figure legend; and (b) provide a link to the terms of use / license information for the base layer image or shapefile. We cannot publish proprietary or copyrighted maps (e.g. Google Maps, Mapquest) and the terms of use for your map base layer must be compatible with our CC BY 4.0 license.

2) Please amend your detailed Financial Disclosure statement. This is published with the article. It must therefore be completed in full sentences and contain the exact wording you wish to be published.

**Reviewers' Comments:**

Reviewer's Responses to Questions

**Key Review Criteria Required for Acceptance?**

**Methods**

-Are the objectives of the study clearly articulated with a clear testable hypothesis stated?

-Is the study design appropriate to address the stated objectives?

-Is the population clearly described and appropriate for the hypothesis being tested?

-Is the sample size sufficient to ensure adequate power to address the hypothesis being tested?

-Were correct statistical analysis used to support conclusions?

-Are there concerns about ethical or regulatory requirements being met?

Reviewer #1: (No Response)

Reviewer #3: The authors drew upon a national avian monitoring system, upon which they could extract data relevant to routine mosquito/pathogen monitoring stations; as well as the eBird database, which has advantages and disadvantages, as mentioned by the authors. The mosquito sampling represents one of the best organized regional mosquito surveillance networks in Europe. They account for sampling biases in questing vs. gravid traps by a clever derivation of their data (“intensity of circulation” = number of years with at least one positive pool).

Reviewer #5: (No Response)

Reviewer #6: The objectives of the study are clearly stated. The study design should be modified to include certain parameters that have not been considered and that could be relevant to the co-modulation of the probability of circulation and transmission dynamics of WNV. It would be advisable to approach the analysis of the data from the study of the association of bird diversity with a focus on the rate of virus circulation rather than the frequency of virus presence over a series of years. The working hypotheses are well presented and the data analysis approach is appropriate, although it could perhaps be somewhat more robust in the authors' first study. There are no ethical implications in the data used by the authors that need to be considered.

**Results**

-Does the analysis presented match the analysis plan?

-Are the results clearly and completely presented?

-Are the figures (Tables, Images) of sufficient quality for clarity?

Reviewer #1: (No Response)

Reviewer #3: The results are clearly presented. See below for recommendations to improve

Reviewer #5: (No Response)

Reviewer #6: The results obtained are presented clearly and in detail, as are the tables and figures.

**Conclusions**

-Are the conclusions supported by the data presented?

-Are the limitations of analysis clearly described?

-Do the authors discuss how these data can be helpful to advance our understanding of the topic under study?

-Is public health relevance addressed?

Reviewer #1: (No Response)

Reviewer #3: For me the discussion required at least three read-throughs to see that the authors addressed all of the concerns that immediately came to my mind upon reading/understanding the data. My suggestions here and below are to clearly provide justification for the analyses, and make sure the connections between the two main analyses are clearly and simply stated. Maybe it is not possible, in which case it is simply an issue of writing style.

Reviewer #5: (No Response)

Reviewer #6: Some of the conclusions are not sufficiently supported by the results due to the lack of robustness of the methodological approach used and the data employed. Some limitations in the data should be taken into account in the interpretation of the authors' findings, and the uncertainty in the findings should be presented in more detail. I believe that the authors should also place more emphasis on the health implications that their results may have, as well as delve deeper into how their results can be translated into actions that can be applied in policy-making to prevent cases of WNF.

**Editorial and Data Presentation Modifications?**

Reviewer #1: (No Response)

Reviewer #3: 1. Figure 3 and accompanying text (sorry - there are no line or page numbers, so it is difficult to specify). please include a measure of goodness-of-fit for these regression results. I recommend a simple R-squared coefficient of determination, also on the figure panels, themselves, under the parameter estimate.

2. Figure 4. I don’t see the “clear spatiotemporal heterogeneity” mentioned in the text. Perhaps the authors could make this clearer (e.g., is there really a difference in richness between a migration-month and nonmigration-month?)

3. Discussion – I recommend referring to the NY99 “strain” more generally as the “North American WNV lineage 1”. However, it raises and important issue – if the authors want to argue that there may be differences between WNV-1 and WNV-2 within the same geographic region (as they do somewhere in the discussion), they may want to re-emphasize whether the positive mosquito pools used in their analyses were WNV-1 or WNV-2 - both circulate in this region of Italy...

Reviewer #5: (No Response)

Reviewer #6: (No Response)

**Summary and General Comments**

Reviewer #1: The manuscript entitled: Association of avian biodiversity and West Nile Virus circulation in Culex mosquitoes in Emilia-Romagna, Italy is a well written and structured, comprehensive and scientifically sound paper. It presents with a clear rationale linking biodiversity and WNV transmission within the dilution effect hypothesis framework. The manuscript benefits from the integration of entomological data from consecutive years along with multiple avian datasets and the use of innovative Bayesian models and rarefaction to control sampling bias, achieving balance of ecological and epidemiological approaches.

General comments for improvement

Language: Some sentences are long and complicated, difficult to follow; shortening them or splitting in 2-3 sentences would improve readability.

Keep focused on the key messages: Τhe authors, in certain points stress too many the methodological details and deviate from the aim of the study especially in the Discussion section, where interpretation of the results is what the reader expects.

Consistency in use of terminology: Always use “avian biodiversity, “instead of “bird diversity,” or “species richness,” DEH after its first use and so on.

Figures and Tables: The Figure references, legends and captions could be slightly simplified to help the readers perceive the key points and results.

Detailed comments

The Abstract section could be shortened by rephrasing, e.g. Instead of “using 11 consecutive years…” say “based on 11 years…”. “Non-passerine species richness was negatively associated with WNV circulation (supporting the DEH), while passerine richness showed a positive association after adjustment.” The last sentence: These findings have important implications…, could end somehow with a stronger effect:…informing biodiversity conservation and integrated public health surveillance activities in Europe.”

The Author Summary section could also benefit with merging or rephrasing, as it repeats content from the abstract and with the addition in the end of a key message.

The Introduction section is comprehensive but rather dense. The authors should consider reducing it by 400-500 words. For example, the first two paragraphs on WNV ecology and vector–host dynamics could be merged to make the manuscript flow for the reader .The 3rd paragraph on biodiversity definitions could be shortened by summarizing the three metric characteristics in one sentence.

The Methods section is detailed and organized, however the authors should consider transferring the long equations and the detailed rarefaction explanations to Supplementary section. Also, please explain and add relevant references regarding the rationale for including a 3 km buffer and cite the mosquito flight range radius somewhere before in the text. Furthermore, when describing INLA–SPDE, a few sentences explaining why, how, this model was chosen and adapted would help the non specialist readers to further understand.

The Results section includes a logical progression to the study’s findings. The authors should ensure that the Figure legends summarize findings, not only a description of content.

E.g. in Figure 3 it should be stated that Higher Shannon and Simpson diversity indices were associated with fewer WNV-positive years, supporting the dilution effect. In addition, the authors should consider presenting in Tables the main regression outputs such as p-values to aid the readers in interpreting. Furthermore, in eBird analysis, the authors should explain briefly why passerine vs non-passerine distinction is important, before presenting the results.

The Discussion section is informative, including prior literature and methodological explanations. However, it is very thorough and it could be shortened without losing robustness and meaning. The paragraphs beginning with “Additionally, our study provides statistical evidence…” and “In terms of environmental factors…” should be closer together in the text, to group mechanistic interpretations. In Discussion the authors should emphasize on novelty, stress that it is the first empirical DEH evidence for WNV in Europe, integration of two avian databases and sources, and 11-year entomological surveillance study. The last three paragraphs starting from To date…could be rewritten aiming to a more concise implications and future work section, including insights for policy and biodiversity conservation, for WNV surveillance and modelling and suggestions for future research.

Minor comments

Abbreviations: Define “DEH” at the first mention in Abstract and again in Introduction for clarity and then use only DEH.

Grammar/ English style recommendations:

Replace “comprehensively covers” with “covers comprehensively”

“analysis were conducted” with “analyses were conducted”

Avoid repetitions of “higher/lower levels of WNV circulation”; use instead “reduced/increased WNV activity.”

Reviewer #3: In this publication, the authors investigate avian biodiversity relative to the presence and relative intensity of WNV circulation. To their advantage, and how this manuscript excels over many others investigating the relationship between biodiversity and pathogen abundance, the authors have a clever sampling design, with excellent scale over a large study area. The search for dilution/amplification effect is never a straight-forward endeavour, and there are many examples of disagreement that stem for differences in methodology. The authors discuss these, and provide adequate discussion to put their analysis within context. However, in general, I think they could do a bit more to justify their use of the models, simply because I like their approach and I wonder how it could be applied to other arbovirus systems.

To summarise their manuscript: For the first part of their study, they categorize their sampling sites according to “WNV intensity” (years of observed positives) in the mosquito population. They then applied these categories to the data from avian monitoring stations to derive a aggregated diversity index per category. They correctly chose to apply multiple estimations of diversity, including rarefaction methods and Chao2 derivations. As they have evidence of a dilution effect, on a rather large temporal scale, they then investigated the relationship between the proportion of positive pools per trap-week, landcover, meteorological parameters, and species richness (divided into passerine and non-passerine species) in a model that accounts for spatial relationships, and incorporates temporal relationships via lagged variables.

The manuscript was really nice to read – the topic is excellent, the approach is well-designed, and the methods are well-established. I admit I have no experience with Bayesian spatiotemporal regression or the INLA-SPDE; however, I have enough background to understand the procedure. The authors did well to devote some of their discussion to the reasoning, benefits, and limitations behind their analyses; and probably the most important sentence from the manuscript “it remains challenging to disentangle whether this reflects a genuine amplifying role…” etc.

My main criticism, is that the authors could add a bit more justification of the spatiotemporal analysis, and the inferences they draw from this. I assume this represents a new application of the INLA-SPDE method, and they should do everything they can to convince us that it can handle the nuances of an arbovirus-vector-host relationship. Namely, I think the assumption that mosquito WNV circulation is a continuous spatially-correlated process across the region needs qualification. I’m not disagreeing, but the evidence for this is indirect and should be provided. It relates specifically to supporting the justification for using “years of activity” as a measure of “WNV intensity”. For example, the largest proportion of sites had three years activity, and those three years – specifically – represented the most intense years of activity across Europe (increased human cases, increased horse cases, increase detections in birds and mosquitoes). It seems contradictory, but is easily explained and the authors should explain it. It is a point worth stressing, because this manuscript could be foundational for other spatiotemporal approaches to understand arbovirus transmission, where vector(s)-vertebrate host(s) relationships are similarly nuanced.

Reviewer #5: (No Response)

Reviewer #6: General comments

The study by Wang et al. seeks to investigate the relationship between the diversity of local bird communities and the circulation dynamics of West Nile virus using a significant set of data from virus surveillance in Emilia-Romagna. The virus surveillance and monitoring programme provides a significant and highly relevant time series of data to deepen on what modulates the infection rate in mosquitoes and the risk of transmission to humans and animals. The study focuses in a very relevant aspect to better understanding WNV ecology and transmission potential, although my impression is that the scarcity of appropriate data on bird diversity encountered by the authors prevents a comprehensive analysis of its effect on the ecological dynamics of the virus and detracts robustness from the findings for the authors' objective.

On the one hand, the authors use data from a national bird survey to derive annual diversity indicators that they relate (in a manner that may be not robust enough for the diversity of elements that can determine local virus circulation in a given year) to the dynamics of WNV infection in mosquito vectors. However, the approach used does not allow for an accurate estimate of the real role of local bird diversity in the dynamics of the virus because the authors are forced to summarise the circulation of the virus in the number of years in which each trap was positive. This estimate of virus circulation frequency is indicative of the probability of virus presence, but does not adequately inform the actual rate of virus circulation in the mosquito population, which is ultimately the relevant finding. A perhaps more appropriate approach for this purpose would be to relate these diversity indicators to the total number of positive Cx. pipiens pools with respect to the total number analysed for each trap and year and in relation to the abundance of Cx. pipiens.

On the other hand, eBird data, even with all the potential biases that they may have and that the authors have tried to minimise, offer the probability of obtaining more accurate spatio-temporal relationships between bird diversity and the infection rate of local mosquito populations. The authors carry out a very well-developed analytical approach, but ultimately they must transform these data into indicators of bird species richness that do not inform about the actual diversity of the local bird community, thus missing the opportunity to identify how variations in local bird communities modulate the infection rate and, with it, the risk of virus transmission.

One aspect that the authors do not consider in either of the two analyses is the local abundance value of Cx. pipiens based on trap capture data, and I believe that this value is extremely relevant in order to identify the extent to which the frequency of virus detection in the mosquitoes analysed or the temporal infection rate are indicators of the relative abundance (which could be called relative density) of infected mosquitoes in the population. Changes in this value may be truly relevant for inferring the effects of bird communities on virus transmission dynamics.

The authors are cautious in their assertions and conclusions, but the findings are actually less robust than the authors consider them to be in order to understand the real effect of the bird community on the dynamics of virus circulation. I think it is important for the authors to provide more depth on the uncertainty associated with their approach in robustly estimating the association between bird diversity and virus dynamics. I think the study is interesting enough to be published, but I believe the authors should make a more concerted effort to clarify the aspects that may not be as robust, such as whether the association between the annual detection frequency of the virus and bird diversity indices supports a dilution effect on transmission as robustly as the authors assert.

The finding of a relationship between passerine and non-passerine richness and the rate of virus circulation is interesting and supports previous results, but it is not a novel finding based on the state of the art of knowledge of the ability of different groups of birds to replicate WNV. Furthermore, it does not provide evidence of which groups within local passerine bird communities may be relevant to the dynamics of virus transmission. It could be that there is a positive correlation between the detectability of passerine species and their abundance that masks an effect that is actually associated with an increase in particular passerine species. A good year from a trophic resource point of view could increase the local attraction of passerines while significantly increasing the abundance of key resident birds hosting Cx. pipiens and WNV.

The authors consider several environmental covariates to study the spatiotemporal association between bird richness and WNV circulation rate, but these covariates do not include any related to the hosts that sustain Cx. pipiens. No variables are included for the abundance of birds or any mammals that could sustain the Cx. pipiens population, even though at the spatial scale of the study, host availability is expected to be one of the most important factors in the population dynamics of a haematophagous arthropod. I believe it is important to explain and justify why these types of covariates are not considered.

Specific comments

It is unclear whether the data used for the period 2013–2023 comes solely from light traps baited with CO₂ or also from gravid traps for the period 2013–August 2017.

Statistical analysis of Farmland Bird index. The authors refer to the parameter estimated to measure the frequency or persistence of WNV circulation as circulation intensity, but given that it is a frequency of detection of the virus's presence in each year, it is not a true intensity value because it does not estimate the number of positive pools in each year, but only the presence/absence of virus detection in each year and each capture area. I suggest that they use the term WNV detection frequency.

I cannot see whether total bird richness was associated with the probability of detecting WNV-positive pools when this parameter is actually more important for assessing the effect of the local bird community as a whole compared to passerine or non-passerine richness. Was the parameter excluded from the selected model? I believe it would enrich the discussion to indicate the role of this global parameter as a joint indicator of richness in the local bird community.

PLOS authors have the option to publish the peer review history of their article (what does this mean? ). If published, this will include your full peer review and any attached files.

**Do you want your identity to be public for this peer review?** For information about this choice, including consent withdrawal, please see our Privacy Policy .

Reviewer #1: No

Reviewer #3: No

Reviewer #5: No

Reviewer #6: No

**Figure resubmission:**
---

## [Decision Letter · Decision Letter 1]

11 Feb 2026

PNTD-D-25-01508R1Association of avian biodiversity and West Nile Virus circulation in *Culex* mosquitoes in Emilia-Romagna, ItalyPLOS Neglected Tropical DiseasesDear Dr. Wang, Thank you for submitting your manuscript to PLOS Neglected Tropical Diseases. After careful consideration, we feel that it has merit but does not fully meet PLOS Neglected Tropical Diseases's publication criteria as it currently stands. Therefore, we invite you to submit a revised version of the manuscript that addresses the points raised during the review process. Please clarify whether your dataset distinguishes between WNV lineages. Both lineages were co-circulating during the study period, and not differentiating between them may affect interpretation given their known phenotypic differences in host and vector competence. This limitation should be explicitly acknowledged, especially for readers who may not be aware of the co-circulation.  Please submit your revised manuscript by Mar 13 2026 11:59PM. If you will need more time than this to complete your revisions, please reply to this message or contact the journal office at plosntds@plos.org.  Please include the following items when submitting your revised manuscript: * A letter that responds to each point raised by the editor and reviewer(s). You should upload this letter as a separate file labeled 'Response to Reviewers '. This file does not need to include responses to any formatting updates and technical items listed in the 'Journal Requirements' section below. * A marked-up copy of your manuscript that highlights changes made to the original version. You should upload this as a separate file labeled 'Revised Manuscript with Track Changes '. * An unmarked version of your revised paper without tracked changes. You should upload this as a separate file labeled 'Manuscript '. If you would like to make changes to your financial disclosure, competing interests statement, or data availability statement, please make these updates within the submission form at the time of resubmission. Guidelines for resubmitting your figure files are available below the reviewer comments at the end of this letter. We look forward to receiving your revised manuscript. Kind regards, Ran Wang, M.D.Academic EditorPLOS Neglected Tropical Diseases Elvina ViennetSection EditorPLOS Neglected Tropical Diseases

Shaden Kamhawi

co-Editor-in-Chief

Paul Brindley

co-Editor-in-Chief

**Reviewers' comments:**  Reviewer's Responses to Questions

**Key Review Criteria Required for Acceptance?**

**Methods**

-Are the objectives of the study clearly articulated with a clear testable hypothesis stated?

-Is the study design appropriate to address the stated objectives?

-Is the population clearly described and appropriate for the hypothesis being tested?

-Is the sample size sufficient to ensure adequate power to address the hypothesis being tested?

-Were correct statistical analysis used to support conclusions?

-Are there concerns about ethical or regulatory requirements being met?

Reviewer #1: (No Response)

Reviewer #3: (No Response)

Reviewer #6: The authors have considered my recommendations of taking into account vector abundace on WNV transmission indices and other methodological considerations.

**Results**

-Does the analysis presented match the analysis plan?

-Are the results clearly and completely presented?

-Are the figures (Tables, Images) of sufficient quality for clarity?

Reviewer #1: (No Response)

Reviewer #3: (No Response)

Reviewer #6: The authors have demonstrated the robustness of their findings, which was one of my main concerns, and clearly stated it in the revised manuscript.

**Conclusions**

-Are the conclusions supported by the data presented?

-Are the limitations of analysis clearly described?

-Do the authors discuss how these data can be helpful to advance our understanding of the topic under study?

-Is public health relevance addressed?

Reviewer #1: (No Response)

Reviewer #3: (No Response)

Reviewer #6: Yes and limitations are clearly presented for a correct interpretation of the conclusions by readers.

**Editorial and Data Presentation Modifications?**

Reviewer #1: (No Response)

Reviewer #3: (No Response)

Reviewer #6: (No Response)

**Summary and General Comments**

Reviewer #1: Τhe comments raised and suggestions proposed at my initial review have been addressed adequately at the revised manuscript.

Reviewer #3: The authors have improved the mansucript according to my suggestions. I think it is a very fine contribution to the understanding of WNV ecology in Europe, and a test of the dilution hypothesis in general. However, I still take issue with one minor point that the authors have not adequately addressed.

From their reply and modification, I understand that they did not have the data to support a test of the effect of WNV lineage on their model. They have acknowledged a comparison with studies in North America, where WNV-1 circulates, in the discussion ("these findings are largely based on North American species and the North American WNV lineage 1 [68], which may not be generalizable to other geographic contexts [69]") and state in their reply "We did not intend to investigate or comment on any potential differences in biodiversity effects across WNV lineages".

However, they must state - explicitly in the manuscript - whether their dataset discriminated between WNV lineages, and note the potential limitations of not differentiating between lineages in their analysis. Both lineages co-circulated in the region, during the time of the study. Phenotypic differences between these viral lineages (in terms of host/vector competences) may obfuscate their findings, and this limitation should be explicitly acknowledged - also potentially for readers who might not be aware that this was the case!

Reviewer #6: (No Response)

PLOS authors have the option to publish the peer review history of their article (what does this mean? ). If published, this will include your full peer review and any attached files.

**Do you want your identity to be public for this peer review?** For information about this choice, including consent withdrawal, please see our Privacy Policy .

Reviewer #1: No

Reviewer #3: No

Reviewer #6: No

**Figure resubmission:**  While revising your submission, we strongly recommend that you use PLOS’s NAAS tool (https://ngplosjournals.pagemajik.ai/artanalysis) to test your figure files. NAAS can convert your figure files to the TIFF file type and meet basic requirements (such as print size, resolution), or provide you with a report on issues that do not meet our requirements and that NAAS cannot fix.

After uploading your figures to PLOS’s NAAS tool - https://ngplosjournals.pagemajik.ai/artanalysis, NAAS will process the files provided and display the results in the "Uploaded Files" section of the page as the processing is complete. If the uploaded figures meet our requirements (or NAAS is able to fix the files to meet our requirements), the figure will be marked as "fixed" above. If NAAS is unable to fix the files, a red "failed" label will appear above. When NAAS has confirmed that the figure files meet our requirements, please download the file via the download option, and include these NAAS processed figure files when submitting your revised manuscript.**Reproducibility:**  To enhance the reproducibility of your results, we recommend that authors of applicable studies deposit laboratory protocols in protocols.io, where a protocol can be assigned its own identifier (DOI) such that it can be cited independently in the future. Additionally, PLOS ONE offers an option to publish peer-reviewed clinical study protocols. Read more information on sharing protocols at https://plos.org/protocols?utm_medium=editorial-email&utm_source=authorletters&utm_campaign=protocols

---

## [Editor Report · Decision Letter 2]

23 Feb 2026

Dear Ms Wang,

We are pleased to inform you that your manuscript 'Association of avian biodiversity and West Nile Virus circulation in *Culex* mosquitoes in Emilia-Romagna, Italy' has been provisionally accepted for publication in PLOS Neglected Tropical Diseases.

Best regards,

Ran Wang, M.D.

Academic Editor

Elvina Viennet

Section Editor

Shaden Kamhawi

co-Editor-in-Chief

Paul Brindley

co-Editor-in-Chief

---

## [Editor Report · Acceptance letter]

Dear Ms Wang,

We are delighted to inform you that your manuscript, "Association of avian biodiversity and West Nile Virus circulation in *Culex* mosquitoes in Emilia-Romagna, Italy," has been formally accepted for publication in PLOS Neglected Tropical Diseases.

Best regards,

Shaden Kamhawi

co-Editor-in-Chief

Paul Brindley

co-Editor-in-Chief
